# Guanidine aptamers are present in vertebrate RNAs associated with calcium signaling and neuromuscular function

Kumari Kavita [1], Aya Narunsky [1], Jessica J. Mohsen [2,3], Isha Mahadeshwar[4], Michael G. Mohsen [1], Yu-Shin Chang [1] & Ronald R. Breaker [1,4] ✉

Guanidine is a protein denaturant that is a widely used constituent in explosives, plastics, and resins. Its effects on muscle contraction were initially reported in 1876, which eventually led to the use of guanidine as a treatment for certain ataxia symptoms such as those caused by Lambert-Eaton disease. However, its mechanisms of therapeutic action remained unknown. Guanidine was recently found to be a widespread natural metabolite through the discovery of four bacterial riboswitch classes that selectively recognize this compound. Here, we report the discovery and biochemical validation of vertebrate members of guanidine-I and -II riboswitch aptamer classes that are associated with numerous genes relevant to neuromuscular function, mostly involved in $Ca^{2+}$ transport or signaling. These findings suggest that guanidine is a widely used signaling molecule that serves as an additional layer of regulation of genes relevant to neuromuscular disorders.

Guanidine, or more specifically its protonated derivative guanidinium ($CH_6N_3^+$; $pK_a = 13.6$)[1], was recognized as a widespread, natural metabolite through the discovery[2–6] of four distinct riboswitch classes in bacteria. These structured RNA devices selectively sense guanidine and regulate genes relevant to its metabolism and transport[7–10]. The ligand-binding aptamer domains for these riboswitch classes, named guanidine-I[2,3], -II[3], -III[4], and IV[5,6], form distinct binding pockets, and crystallographic data[11–14] have shown that representatives from guanidine-I, -II, and -III use hydrogen bonding and cation-π interactions to selectively recognize guanidine. Other common biological metabolites that carry a guanidinium moiety (e.g., arginine) are expected to be strongly rejected, as are other synthetic analogs[2–6]. These findings support the conclusion that many bacteria are naturally exposed to guanidine and must make gene expression changes to adapt to its presence.

Because guanidine riboswitches are associated with the messenger RNAs (mRNAs) of protein-coding genes, an accurate list of bacterial proteins relevant to guanidine biology can be generated. For example, genes commonly associated with guanidine-sensing riboswitches and previously annotated as encoding urea carboxylase enzymes have proven instead to code for guanidine carboxylases[2]. Similarly, some associated genes generally annotated as small multi-drug resistance (SMR) transporters have proven to be selective transporters of guanidine[2,15,16]. These initial findings inspired additional studies revealing the extensive involvement of guanidine in various biological processes. For example, some plant species produce large amounts of guanidine[17], and many bacterial species manipulate its cellular concentration[2] or otherwise exploit guanidine as a source of fixed nitrogen[10,18,19]. Also, the location of guanidine riboswitches upstream of a previously unknown gene class enabled researchers to identify the associated gene products as nickel-dependent guanidine hydrolases[20,21].

Even though guanidine is known to be made by certain bacterial and plant enzymes[17,22–26], there have been no previous reports that it is naturally produced as a metabolite in any animal species. However, guanidine has been known since 1876 to affect muscle function[27–30]. Furthermore, guanidine has been used[31,32] as a therapeutic agent in humans for the treatment of Lambert-Eaton myasthenic syndrome, a

[1]Department of Molecular, Cellular and Developmental Biology, Yale University, New Haven, CT 06511, USA. [2]Department of Chemistry, Yale University, New Haven, CT 06511, USA. [3]Institute for Biomolecular Design and Discovery, Yale University, West Haven, CT 06516, USA. [4]Department of Molecular Biophysics and Biochemistry, Yale University, New Haven, CT 06511, USA. ✉e-mail: ronald.breaker@yale.edu

 

neuromuscular disorder wherein nerve cells fail to signal muscle movement[33]. This disorder is caused by an autoimmune response against divalent calcium ($Ca^{2+}$) transporters. Similarly, guanidine can help overcome the effects of the paralytic neurotoxin curare[29] and botulism[34], and has been examined for treatment of other neuromuscular disorders[35–37]. These previously reported effects of guanidine in humans and other animals have long hinted that vertebrates carry either natural or fortuitous binding sites for guanidine, or perhaps for one or more of its metabolic derivatives.

In the present study, we use computational methods based on comparative nucleotide sequence analysis[38,39] to search eukaryotic genomes for RNA domains that closely correspond to any of the four guanidine riboswitch classes from bacteria[2–6]. Dozens of RNA domains are uncovered in vertebrates with sequence and structural features similar to the consensus models of either guanidine-I or guanidine-II riboswitch aptamers from bacteria. These bioinformatic hits are often present, either in sense or antisense orientations, in the noncoding portions of mRNAs for genes relevant to $Ca^{2+}$ homeostasis/signaling or to neuromuscular function. Given the known relationship between guanidine exposure and neuromuscular function, we conclude that these vertebrate RNA domains likely represent components of guanidine riboswitches.

To assess the hypothesis that vertebrates make extensive use of guanidine riboswitches, we conduct assays to evaluate the ligand-binding functions of representative guanidine-II riboswitch candidates. Although some of these RNAs have unusual sequence or structural features compared to their bacterial counterparts, vertebrate guanidine-II candidates exhibit robust and selective binding to guanidine. In addition, a vertebrate protein whose mRNA is associated with a guanidine-II aptamer alters guanidine concentration when expressed in bacterial cells. Finally, the list of gene associations reveals a complex network of proteins that appear to be integrally involved in $Ca^{2+}$ signaling, neuromuscular function, and, in rare instances, nitric oxide production. These and other findings support the hypothesis that many vertebrate species naturally make productive use of guanidine, either directly or indirectly, to affect $Ca^{2+}$ biology and neuromuscular activities. Our findings also support the long-standing hypothesis that animals use metabolite-binding riboswitches to regulate the expression of genes. Thus, riboswitch mechanisms for molecular sensing and gene control are likely to be far more widespread among eukaryotic species than is currently known.

## Results

### Guanidine-II riboswitch aptamer candidates in vertebrates

More than 60 distinct riboswitch classes from bacteria have been experimentally validated[40–42], but only riboswitches for the enzyme cofactor thiamin pyrophosphate (TPP) have been convincingly demonstrated to exist in eukaryotes[43,44] - primarily in fungi[45–48] and plants[49–51]. Although claims of additional eukaryotic classes have been made, reasons why these reports are likely to be incorrect have been described elsewhere[42]. Various technical challenges have hampered past efforts to discover and confirm examples of riboswitches in vertebrates. For example, eukaryotic genomes are orders of magnitude larger than bacterial genomes, which makes it more difficult to use biochemical, genetic, or computational approaches to find previously undiscovered riboswitch classes in these species.

A powerful approach to uncovering eukaryotic riboswitch candidates involves the use of computational algorithms[52] to identify matches to (or close variants[53,54] of) the consensus sequences and structural models for known bacterial riboswitch aptamer classes. Searching for additional representatives of a known class is less computationally demanding than searching for members of a previously undiscovered RNA aptamer architecture. In addition, the process of assessing the validity of eukaryotic riboswitch candidates can benefit from existing knowledge about the structure and function of members of the original bacterial riboswitch class. In response to the expanding understanding of the roles of guanidine in biology[2–26] and recognizing the inherent utility of computational searches for riboswitch representatives based on comparative sequence analysis algorithms[52–54], we conducted searches of eukaryotic genomes for close matches to the four classes of bacterial guanidine riboswitches[2–6].

We readily identified several strong matches to the bacterial guanidine-II aptamer[3] consensus (Fig. 1a) (previously called mini-$ykkC$ motif[55]) associated with the carbonic anhydrase 8 ($CA8$) gene of several species of vertebrates, including elephants (Fig. 1b), parrots, and certain fish species (Supplementary Fig. 1). Mutations in the human $CA8$ gene (see below for further discussion regarding CA8 proteins) are known to cause spinocerebellar ataxia[56–58], which is a neuromuscular disorder that causes a loss of balance, coordination, and muscle control. Most of these aptamer candidates reside in a region that corresponds to an intron, often in an antisense orientation, of the $CA8$ mRNA. Candidates in other species were also observed in this same region that carry imperfections compared to the bacterial consensus model or that lack one of the two hairpins characteristic of bacterial guanidine-II riboswitch aptamers.

After conducting our search, we noticed that some of these same hits are also present in a list of candidate guanidine-II aptamers in the Rfam database of RNA sequence families[59]. Over 100 examples are listed that range throughout the eukaryotic domain of life. Many of these additional candidates strongly correspond to the guanidine-II aptamer consensus and likewise have intriguing gene associations (Supplementary Fig. 2), such as candidates associated with the $PISD$ gene of the double-banded courser (*Rhinoptilus africanus*) (Fig. 1c) and other birds, as well as the $CACNA1C$ gene of the common bottlenose

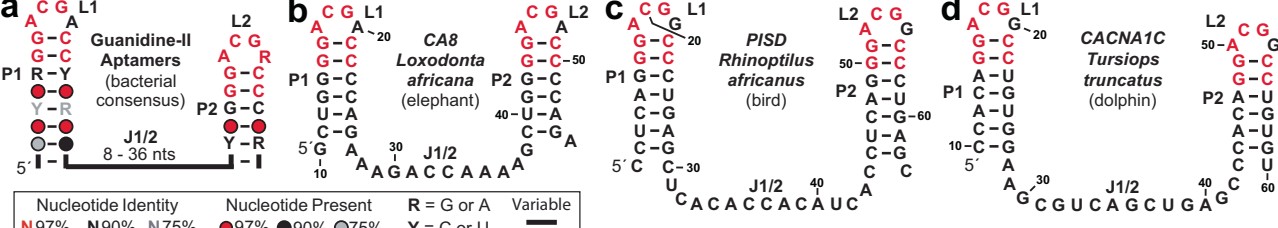

**Fig. 1 | Sequence and secondary structure models for the consensus bacterial guanidine-II riboswitch class and representative vertebrate hits. a** Updated consensus sequence and structural model for bacterial guanidine-II riboswitches. The consensus was updated from previous models[3,55] by examining 1977 distinct bacterial representatives. **b** Sequence and secondary structure model for a guanidine-II riboswitch aptamer candidate from the $CA8$ gene of African elephant (*Loxodonta africana*). **c** Sequence and secondary structure model for a guanidine-II riboswitch aptamer candidate associated with the $PISD$ gene from the bird *Rhinoptilus africanus*. **d** Sequence and secondary structure model for a guanidine-II riboswitch aptamer candidate associated with the $CACNA1C$ gene of common bottlenose dolphin (*Tursiops truncatus*). Note: Additional hairpins conforming to the guanidine-II aptamer consensus reside near to the examples presented here. Nucleotide numbering for each panel is based on the RNA constructs used for ligand binding assays.

dolphin (*Tursiops truncatus*) (Fig. 1d) and other marine mammals. Intriguingly, the protein encoded by the *CACNA1C* gene is a calcium voltage-gated channel subunit Alpha1 C[60]. Mutations in the human gene are relevant to neuromuscular, skeletal, and heart (long Qt syndrome, Brugada syndrome) defects[60,61]. Characteristics of some of these additional gene associations are discussed in more detail later.

## A *CA8*-associated guanidine-II aptamer candidate binds guanidine

Bacterial guanidine-II riboswitch aptamers are formed by short, tandem hairpins that function cooperatively to bind two guanidine ligands[3,12,13,62–64]. Likewise, biochemical evidence described herein indicates that mammalian candidates also bind two guanidine molecules in a cooperative fashion. An RNA structure analysis assay called in-line probing[65,66] was used to determine whether RNA constructs encompassing guanidine-II aptamer candidates from eukaryotes indeed bind guanidine. In-line probing exploits the fact that spontaneous scission of RNA phosphodiester linkages is accelerated in structurally unconstrained regions compared to highly structured regions[65,67]. If an RNA undergoes structural modulation upon ligand binding, then the pattern of spontaneous RNA cleavage products generated during in-line probing reactions will change accordingly.

An RNA construct (Fig. 2a) called 62 *CA8*, encompassing a 62-nucleotide region antisense to an intronic region of the *CA8* pre-mRNA of *Loxodonta africana*, was prepared by in vitro transcription, 5′ $^{32}$P-radiolabeled, and subjected to in-line probing in the presence of various concentrations of guanidine. Note that this construct includes only the first two hairpins that fully match the bacterial consensus, from among nine near-identical hairpins clustered in this region (Supplementary Fig. 1, hairpins 2 and 3 or H2 and H3). The spontaneous RNA cleavage products (Fig. 2b) reveal robust binding to

guanidine with characteristics consistent with 2-to-1 guanidine-to-RNA binding (Hill coefficient of 1.46) with an apparent dissociation constant ($K_D$) of ~115 μM (Fig. 2c). From a total of three replicate experiments (Supplementary Fig. 3), the average Hill coefficient is 1.50 ± 0.3 and the average $K_D$ is 111 ± 3 μM.

Furthermore, RNA product bands corresponding to strand scission at positions 20 and 48 (Fig. 2a) in the highly conserved ligand-binding loops of H2 and H3 are similar in intensity (Fig. 2b) to that observed for the equivalent positions in bacterial examples[3]. This is likely because the two A nucleotides at these positions stack on each other by projecting outward compared to the rest of the loop nucleotides[12,13], thereby providing a greater probability of adopting an in-line configuration compared to other parts of the conserved substructures. All these features conform to the known structural[12,13] and ligand-binding[3] characteristics for bacterial guanidine-II riboswitch aptamers.

The 62 *CA8* construct also exhibits a pattern of ligand binding selectivity similar to that observed for bacterial representatives[3]. For example, guanidine derivatives carrying small additions such as a methyl or amino group exhibit binding affinities that are similar to that for guanidine, whereas larger additions reduce binding affinity for the analog (Supplementary Fig. 4). In addition, a mutation to the highly conserved C nucleotide at nucleotide position 18 (C18U) eliminates binding by H2 and greatly diminishes binding affinity at H3 as evaluated by in-line probing assays conducted with 1 mM guanidine (Supplementary Fig. 5).

## A cluster of guanidine-II hairpins exhibits complex functions

Although all the binding characteristics of the 62 *CA8* RNA construct described above are consistent with the known properties of guanidine-II aptamer function, the simplicity of the consensus

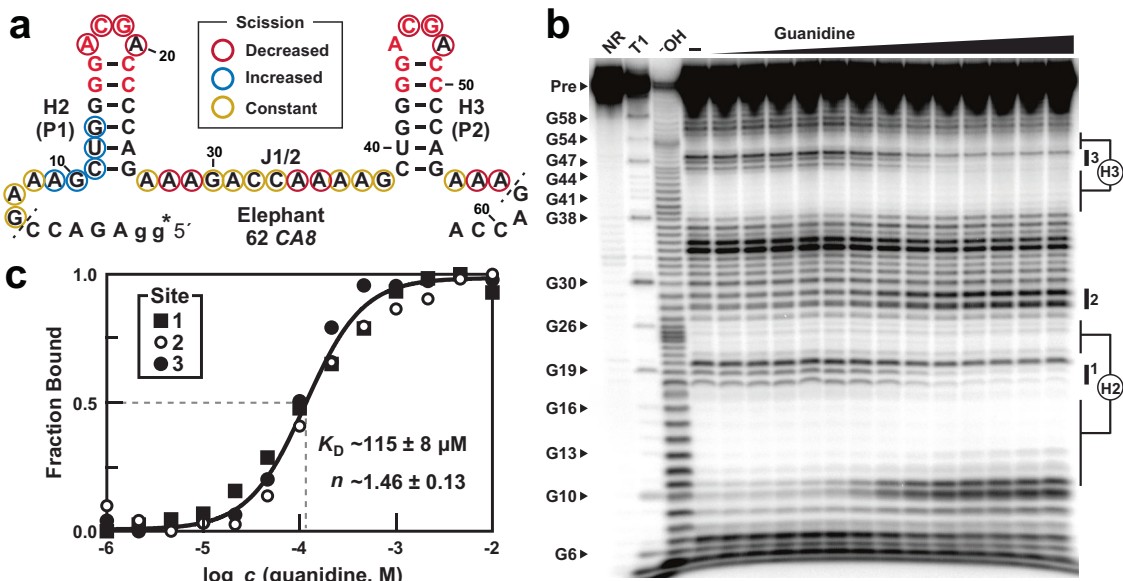

**Fig. 2 | Ligand binding by the guanidine-II aptamer candidate associated with the elephant *CA8* gene. a** Sequence and secondary structure for the African elephant representative 62 *CA8*, encompassing 62 nucleotides from the *CA8* mRNA antisense plus two additional guanosine nucleotides (gg) added to the 5′ end to support production by in vitro transcription. The asterisk identifies the site of the $^{32}$P-radiolabel. **b** Representative image of the polyacrylamide gel electrophoresis (PAGE) separation of the products of in-line probing reactions using the elephant *CA8* aptamer. See Supplementary Fig. 3 for two additional replicates. In-line probing reactions were conducted in the absence of ligand (–), or in the presence of guanidine ranging from 1 μM to 10 mM. NR, T1, and –OH indicate precursor RNAs subjected to no reaction, partial digestion with RNase T1 (cleaves after G

nucleotides), and partial digestion at elevated pH, respectively. Pre indicates the band corresponding to the precursor (full-length) 5′ $^{32}$P-labeled RNA. Some bands corresponding to RNase T1 digest are indicated with nucleotide numbers to assist in mapping sites of spontaneous cleavage. Nucleotides involved in forming base-paired stems P1 and P2 are indicated on the right. Bands undergoing substantial change in intensity are designated as sites 1, 2, and 3 (see vertical bars). **c** Plot of the fraction of RNA bound to ligand versus the logarithm of the molar concentration of guanidine. Fraction bound values are derived from the band intensity changes at sites 1, 2 and 3 as depicted in b. The error values are the standard error of the mean determined by goodness of fit to a sigmoidal curve. See Supplementary Fig. 3 for two additional replicates.

sequence and secondary structure model for this aptamer class could mean that false positive hits identified by computational searches might also have a high propensity to exhibit aptamer function even if they are not components of a natural riboswitch. However, it seems unlikely that natural RNAs, by simple chance, would have more than one or two hairpins that conform to the guanidine-II aptamer consensus unless they had biological utility. Intriguingly, we found that an RNA construct (called 112 *CA8*) carrying 112 nucleotides also from the African elephant *CA8* gene, encompassing hairpins 4 through 7 (H4-H7) (Supplementary Fig. 2, Supplementary Fig. 6a), exhibits additional unusual characteristics.

In-line probing of the wild-type (WT) 112 *CA8* construct revealed that hairpin H4, which naturally carries the disabling C-to-U mutation noted above, as expected, fails to bind guanidine under the conditions examined. Surprisingly, the loops of H5, H6, and H7 all undergo structural modulation when guanidine concentrations are elevated (Supplementary Fig. 6b), indicating ligand binding at these three sites. This was unexpected because the cooperative binding of guanidine by pairwise interactions between hairpins would mean that only two hairpins should form such a partnership and bind two guanidine molecules. However, three hairpins presumably bind three guanidine molecules, suggesting that there is another way for an additional hairpin to strongly bind guanidine without forming the typical consecutive Watson-Crick base pairs with a partner hairpin loop that matches the consensus sequence.

Furthermore, the apparent $K_D$ values for guanidine binding at H5, H6, and H7 (less than $5\,\mu M$) (Supplementary Fig. 6b) appear to be substantially better than that of the construct carrying only H2 and H3 (~$110\,\mu M$) (Fig. 2, Supplementary Fig. 3) or for examples of bacterial guanidine-II aptamers (~$60\,\mu M$) reported previously[3]. This hints at the intriguing possibility that the nine hairpins, which mimic guanidine-II binding aptamers, form structures that enable cooperative ligand binding between two sets of hairpin dimers, in addition to the local cooperative interactions resulting from each pairwise hairpin partnership. Further evidence for this hypothesis is observed when a mutant version of the 112 *CA8* construct carrying a single C-to-U mutation in the loop of H5 is examined by in-line probing. This construct, called M1, fails to exhibit evidence of guanidine binding by H4 or by the mutated H5, but retains guanidine-induced structure modulation in the loops of H6 and H7 (Supplementary Fig. 7). These results suggest that H6 and H7 partner to form the classic aptamer structure that binds two guanidine molecules cooperatively. We speculate that the naturally mutated H4 and WT H5 normally partner, perhaps using a U•G wobble pair and a G-C base pair to permit guanidine binding only by the loop of H5. When the loop of H5 is mutated, as in construct M1, binding in H5 by the H4-H5 partnership is disrupted. With the loss of guanidine binding by H5, cooperative improvement of ligand binding affinity by the H6-H7 partnership is also precluded. This is evident by the fact that the $K_D$ values for guanidine binding by H6 and H7 are substantially poorer in the M1 construct (~$20\,\mu M$) compared to WT (Supplementary Figs. 7, 8).

Although additional experiments will be needed to establish the structural features and characteristics of the larger stretch of nine hairpins found in the African elephant, the complexity described herein strongly indicates that there is an intricate interplay between guanidine and the antisense RNA transcript of the *CA8* gene in this species. This complexity is unlikely to occur in an RNA that is a bioinformatic false positive, but rather these characteristics are consistent with the hypothesis that most guanidine-II aptamer candidates in vertebrates are biologically functional.

### Human CA8 protein increases guanidine when expressed in bacteria

The association of guanidine-II aptamers with *CA8* genes suggests that the activity of CA8 proteins, also called CA-VIII or carbonic anhydrase

related protein (CARP)[68–73], might instead be relevant to guanidine metabolism. Guanidine riboswitches in bacteria regulate the expression of proteins that transport or metabolize this same ligand. CA8 proteins have no previously assigned enzymatic function despite their similarity to well-established carbonic anhydrase enzymes[69–71]. Expression of CA8 is most abundant in Purkinje cells[69,72,73], which are involved in neuronal signaling of muscle activity. Intriguingly, disabling mutations in CA8 cause spinocerebellar ataxia in humans[56–58], which is a neuromuscular disorder involving disrupted $Ca^{2+}$ signaling that in some instances results in quadrupedal locomotion in patients[56]. As noted above, guanidine has previously been used to treat certain neuromuscular disorders[32–37].

One intriguing possibility is that CA8 catalyzes a reaction analogous to the decarboxylation of water (carbonic anhydrase activity), such as the decarboxylation of carboxyguanidine (carboxyguanidine decarboxylase). Note that carboxyguanidine is already known to be commonly produced by bacteria to avoid the toxic effects of guanidine[2], or as part of a pathway for guanidine degradation[19]. To assess whether CA8 influences guanidine concentration in cells, the human *CA8* gene was expressed in a strain of the bacterium *Bacillus subtilis* that was previously[2] adapted to carry a guanidine-responsive reporter gene system. Specifically, a bacterial guanidine-I riboswitch has been fused to the *Escherichia coli* β-galactosidase (*lacZ*) gene to yield increased reporter gene expression when guanidine concentrations are elevated. This riboswitch-reporter fusion construct is ideal for the current study because the guanidine-I riboswitch aptamer is highly selective for guanidine[2] and thus is expected to reject even close metabolite analogs such as arginine, agmatine, creatine, or others that naturally carry a guanidine or guanidine-like moiety. Thus, reporter gene expression, as measured by the blue color generated when β-galactosidase cleaves the indicator molecule X-gal, should be proportional only to cellular guanidine concentrations.

As observed previously[2], when WT *B. subtilis* cells are grown on solid (Luria-Bertani, LB) agar media, the riboswitch-reporter fusion construct is in the OFF state. However, the addition of a solution of 6 M guanidine to a cellulose filter disk on inoculated plates grown overnight (agar diffusion assay) results in a small no-growth zone caused by guanidine toxicity. Farther from the filter disk is a visually evident halo of blue color (ON state) indicating that a high but sublethal concentration of guanidine is present (Fig. 3a, left). Intriguingly, plates inoculated with *B. subtilis* cells carrying the human *CA8* gene (+*CA8*) exhibit a larger blue halo when subjected to the same amount of guanidine in the agar diffusion assay (Fig. 3a, right). This visual effect is also apparent when photos of the culture plates are quantitated (Fig. 3b), supporting the hypothesis that the human CA8 protein increases cellular guanidine concentrations.

To further evaluate the effects of CA8 on guanidine concentrations, we conducted riboswitch-reporter assays using liquid LB media. WT and +*CA8* cells gave similar, low expression in the absence of added guanidine, and when $300\,\mu M$ guanidine was added to the media (Fig. 4a). The low reporter gene expression exhibited by *B. subtilis* cells is expected due to the action of the guanidine transporter system naturally encoded by the *ykkCD* operon in this species[2]. This transporter is also likely to keep pace with any modest increases in guanidine production that might be caused by the presence of human CA8. We previously[2] determined that deletion of the *ykkCD* operon (Δ*ykkCD*) causes *B. subtilis* cells to become more sensitive to guanidine, and that these knock-out cells yield higher reporter gene expression from the riboswitch-reporter fusion construct used in the present study. With consideration of these characteristics, we reevaluated the effects of guanidine exposure on reporter gene expression for both the Δ*ykkCD* strain and this same strain carrying the human *CA8* gene (Δ*ykkCD*/+*CA8*).

As observed previously[2], Δ*ykkCD* cells exhibit higher reporter gene expression (~25-fold increase) compared to WT cells when

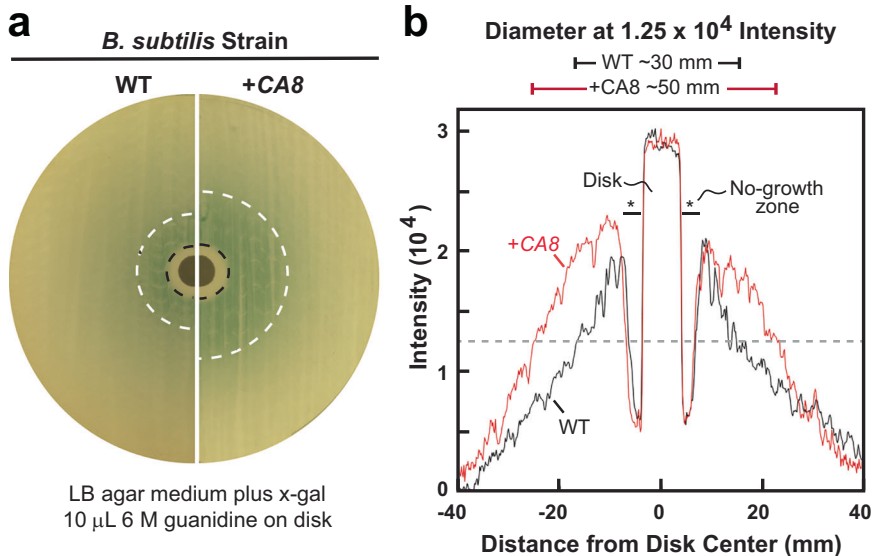

**Fig. 3 | Evidence that human CA8 protein increases cellular guanidine concentration in bacteria. a** Images of agar diffusion assays using LB media containing X-gal and supplemented by application of 10 μM 6 M guanidine on a filter disk (center of plate). Images of plates cultured with WT or +*CA8* strains of *B. subtilis* were recorded after overnight growth at 37 °C, and hemispheres of the two types of plates are depicted adjacently for comparison. Visually apparent transition boundaries for no-growth to growth (black dashed lines) and high β-galactosidase reporter gene expression (blue color) to low expression (white dashed lines) are identified. **b** Plot of the pixel intensity versus distance from the center of the filter disk along horizontal lines dissecting the gray-scale images of the full agar diffusion assay plates in a. Diameters representing high-level reporter gene expression halos (above an intensity of 1.25 ×10⁴ units, gray dashed line) are indicated at the top.

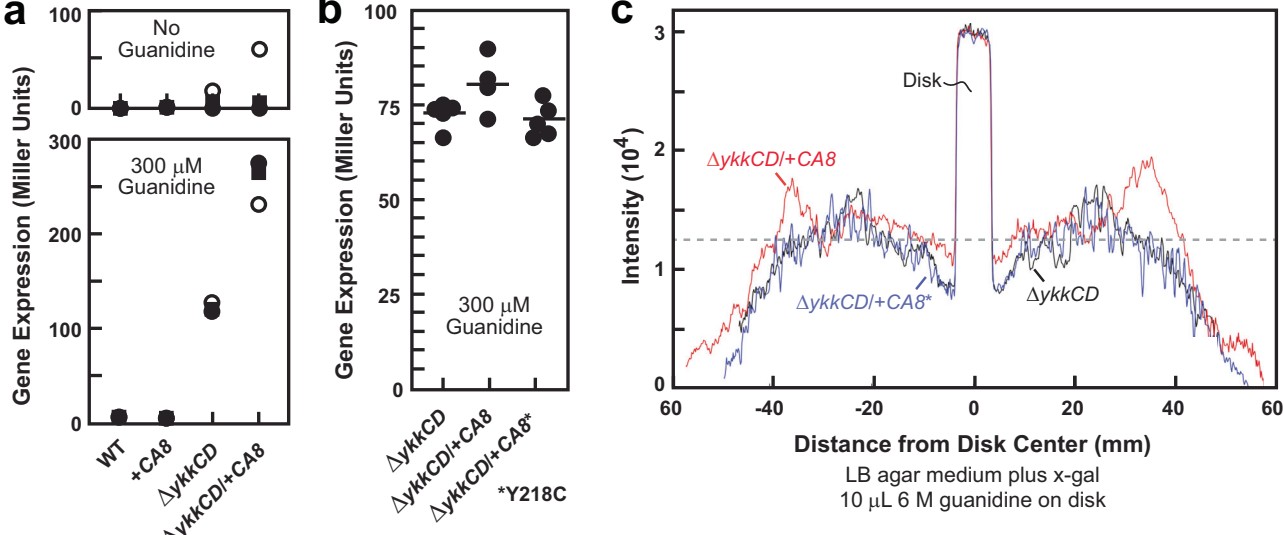

**Fig. 4 | Evidence that mutant human CA8 protein fails to increase cellular guanidine concentration in bacteria. a** Plot of β-galactosidase reporter gene expression (Miller units) for liquid LB media assays in the absence or presence of 300 μM guanidine. Data points represent three experimental replicates for each of the four *B. subtilis* strains indicated. **b** Plot of β-galactosidase reporter gene expression as described for a wherein cultures were allowed to grow to near equal $OD_{600}$ prior to conducting Miller assays. The asterisk indicates the Y218C mutant of CA8 was used. **c** Plots as depicted in Fig. 3b for agar diffusion assays using the strains indicated.

exposed to guanidine (Fig. 4a), consistent with the inability of Δ*ykkCD* cells to expel this molecule. Importantly, the Δ*ykkCD*/+*CA8* strain exhibits the highest level of reporter gene expression observed (~50-fold increase over WT), which is again consistent with the conclusion that human CA8 modestly elevates cellular guanidine concentrations and that this becomes detectable in liquid LB cultures supplemented with 300 μM guanidine when guanidine export is disabled. Importantly, this small increase in reporter gene expression brought about by WT CA8 protein is not observed when a mutant form of CA8 is expressed that carries a Y218C mutation (Fig. 4b). This mutant protein was chosen for evaluation because it is known to be a cause of spinocerebellar ataxia in humans[74], which is an outcome of disabling mutations in *CA8*. Similar results are observed in agar diffusion assays (Fig. 4c), again demonstrating that the WT CA8 protein modestly increases guanidine-induced reporter gene expression, whereas the mutant protein does not.

One possible mechanism for producing these results is that *B. subtilis* modifies excess guanidine, for example, to reduce its toxicity (such as by carboxylation as seen with other bacterial species[2]), and that CA8 proteins reverse this modification, thereby increasing the

concentration of free guanidine. Unfortunately, carboxyguanidine is highly unstable[19] and cannot be obtained to assess whether CA8 functions as a guanidine decarboxylase. Another possibility is that CA8 generates guanidine by acting on a natural metabolite that carries a guanidyl moiety, such as arginine or creatine, but uncertainty over possible substrates and reaction conditions complicates any experimental evaluation of this hypothesis. Given these technical challenges, we sought further evidence for guanidine riboswitches in vertebrates using other approaches.

### Other guanidine-II aptamers associate with Ca²⁺ biology genes

To further search for evidence that guanidine-II riboswitch aptamers are contributing more broadly to natural guanidine signaling in vertebrates, additional hits (Supplementary Fig. 2) in the Rfam database[59] were evaluated using bioinformatic methods. Each of these candidates must be interpreted with caution because some past eukaryotic riboswitch candidates reported by others have proven to represent bacterial sequence contamination or false-positive hits due to the simplicity of the consensus model (unpublished findings). Furthermore, some of these additional hits that match the guanidine-II consensus model are not associated with an annotated gene.

Intriguingly, when a hit clearly resides near or within a eukaryotic gene, the function of the gene's protein product is often relevant to Ca²⁺ biology, neuromuscular disorders, or bone development diseases (Supplementary Fig. 2). For example, the gene most associated with vertebrate guanidine-II aptamer candidates from Rfam is annotated as *PISD*, which codes for phosphatidylserine decarboxylase that catalyzes the production of phosphatidylethanolamine[75]. Genetic defects in the human *PISD* gene are the cause of Liberfarb Syndrome[76], which results in bone and neurological development problems. A total of ten distinct representatives from fish and bird species (Supplementary Fig. 9) are located ~10 kb upstream and in the same orientation of *PISD* mRNAs. Intriguingly, many bacterial riboswitches[40–42] and some eukaryotic TPP riboswitches[45,46] reside in the 5′-untranslated regions of mRNAs, although it is not known if the guanidine-II candidates are part of the *PISD* transcript. Regardless, an RNA construct encompassing the two hairpins of the *PISD*-associated candidate from *Rhinoptilus africanus* (bird) exhibits an approximate $K_D$ of 1 mM for guanidine (Supplementary Fig. 10).

Another notable candidate guanidine-II aptamer system is associated with the *CACNA1C* gene of diverse vertebrate species. As stated earlier, the protein derived from this gene forms a voltage-gated calcium ion channel whose disruption causes neuromuscular, skeletal, and heart function defects[60,61]. The common bottlenose dolphin (*Tursiops truncates*) and orca (*Orcinus orca*) carry a tandem hairpin arrangement (Fig. 1d) antisense to the *CACNA1C* gene that matches the consensus for bacterial guanidine-II riboswitch aptamers. Each species carries an additional consensus hairpin plus one or two that violate the consensus features (Supplementary Fig. 11). These variant hairpins could be non-functional, or they might partner with an adjacent, consensus hairpin to help form a single ligand binding pocket. In-line probing analysis of an antisense *CACNA1C* RNA construct carrying two consensus hairpins indeed bind guanidine (Supplementary Fig. 12), although the apparent affinity for this construct is much poorer than that observed for other candidates – perhaps due to the absence of flanking nucleotides that provide necessary structural context. Regardless, these findings again indicate that candidate guanidine-II aptamer systems are receptors for guanidine that associate with numerous genes relevant to Ca²⁺ signaling and neuromuscular functions.

### Vertebrate RNAs resemble the bacterial guanidine-I aptamer class

Given the abundance of vertebrate RNAs that closely conform to the guanidine-II aptamer class, we looked more exhaustively for RNAs in these species that are similar to the consensus models for other known classes of guanidine aptamers. A total of 89 hits (e-value less than 10) similar to the bacterial guanidine-I aptamer class (Fig. 5a) were identified that are associated with genes broadly relevant to neuromuscular function and Ca²⁺ signaling (Fig. 5b, Supplementary Fig. 13). Genes with the most initial hits (*i.e.*, before iterative searching using refined consensus models) among vertebrates include *ITPR1* (12 hits), *MAGI1* (9), *ATM* (5), *WWOX* (4), *BAZ1B* (3), and *STRN* (3). An RNA consensus model (Fig. 5c) based on an expanded collection of vertebrate *ITPR1* candidates is further discussed below.

Perhaps most noteworthy from this list of associated genes is *ITPR1* (inositol 1,4,5-triphosphate receptor type 1), whose protein product plays a role in cellular Ca²⁺ control[77]. Intriguingly, ITPR1 protein binding is the only established function of CA8 proteins[78], whose genes in some species carry sequences and structures matching guanidine-II riboswitch aptamers as described above. Mutations in ITPR1 are also known to cause spinocerebellar ataxia[79], like that observed for mutations to CA8. Thus, candidate guanidine riboswitch aptamers from two different classes map to genes coding for interacting proteins whose disruption both cause the same neuromuscular disorder.

Additional bioinformatic analyses reveal that most reptilian and mammalian species carry an RNA sequence associated with their *ITPR1* gene similar to the initial 12 hits noted above (Fig. 6). This distribution of RNA structures similar to guanidine-I aptamers is consistent with the hypothesis that the vast majority of reptiles and mammals use guanidine to affect the structure and function of *ITPR1* transcripts. A sequence alignment of over 50 hits from mammalian *ITPR1* genes (Supplementary Fig. 14) exhibit extensive similarity to the 3′-most nucleotides of the bacterial guanidine-I consensus model[2]. This substructure of the bacterial aptamer forms most of the guanidine binding pocket, including regions that carry three of the four nucleotides that directly contact guanidine[11]. The bacterial (Fig. 5a) and vertebrate (Fig. 5c) RNA consensus models differ mostly in their 5′ regions, which in bacteria is known to form a structural cradle into which the ligand binding site docks[11]. It is possible that mammalian RNAs with similarities to guanidine-I aptamers of bacteria use a different cradle structure to support the ligand binding pocket. Initial analysis of a human *ITPR1* RNA construct by in-line probing revealed that the RNA is poorly folded under our assay conditions. Thus, assessment of its ligand binding characteristics will require the identification of constructs that exhibit improved folding.

Other genes associated with guanidine-I aptamer candidates expand the collection of genes with known links to neuronal development or function and/or Ca²⁺ signaling. MAGI1 (membrane-associated guanylate kinase) is involved in diverse cellular functions, including the formation of neuronal synapses[80]. Evidence also indicates that the protein makes direct interactions with at least one Ca²⁺-activated ion channel[81], which is consistent with our hypothesis that guanidine is linked to vertebrate neuromuscular function and Ca²⁺ signaling. ATM (ataxia-telangiectasia-mutated) protein is linked to neuromuscular function and Ca²⁺ signaling[82,83]. WWOX (WW domain-containing oxidoreductase) protein is also linked to neuromuscular function, including spinocerebellar ataxia[84], and Ca²⁺ signaling[85]. This striking association of guanidine-I and guanidine-II aptamer candidates with genes relevant to Ca²⁺ signaling and neuromuscular function is unlikely to be coincidental.

## Discussion

Bioinformatic data reported herein reveal that guanidine-I and guanidine-II riboswitch aptamer candidates are widespread among vertebrates, and that they associate with genes relevant to Ca²⁺ signaling and neuromuscular function. Vertebrate guanidine-II aptamer function is confirmed by ligand binding assays with several representatives that fold to form the consensus P1 and P2 hairpin structures.

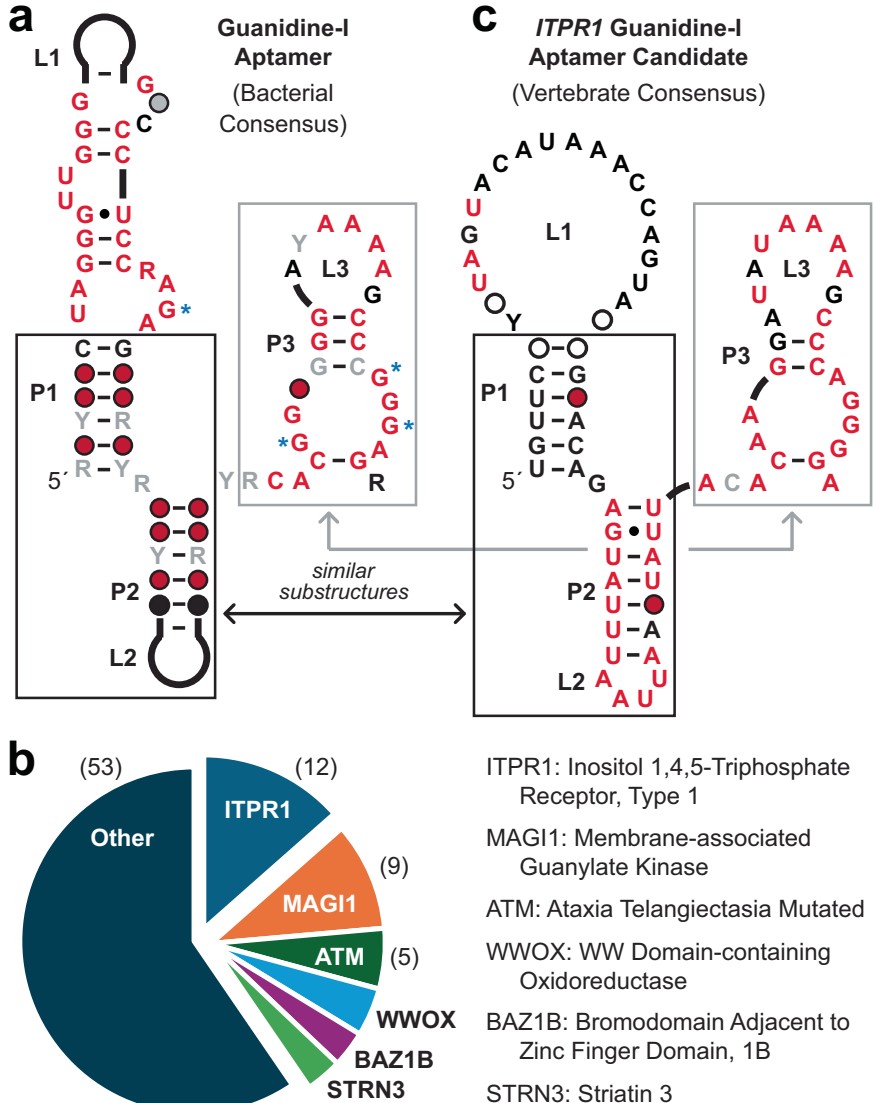

**Fig. 5 | Conserved RNA domains similar to bacterial guanidine-I riboswitch aptamers are associated with numerous vertebrate genes relevant to Ca²⁺ signaling and neuromuscular function. a** Consensus sequence and structural model for bacterial guanidine-I riboswitch aptamers[2] used as the basis for the search for similar RNAs in vertebrates. Annotations are as described for Fig. 1. **b** Pie chart depicting genes associated with RNAs similar to bacterial guanidine-I aptamers. For the most commonly associated genes, the number of hits identified by an initial comparative sequence analysis run using an e-value cutoff of 10 are given in parentheses. **c** Consensus sequence and structural model for the top 51 hits identified in vertebrate *ITPR1* mRNAs (Supplementary Fig. 14) using a comparative sequence analysis search pipeline (see Materials and Methods). Boxed regions identify sequences and structural features that are similar between the bacterial and vertebrate models.

ITPR1: Inositol 1,4,5-Triphosphate Receptor, Type 1

MAGI1: Membrane-associated Guanylate Kinase

ATM: Ataxia Telangiectasia Mutated

WWOX: WW Domain-containing Oxidoreductase

BAZ1B: Bromodomain Adjacent to Zinc Finger Domain, 1B

STRN3: Striatin 3

These constructs cooperatively bind two guanidine molecules and exhibit ligand affinities similar to those observed for their bacterial counterparts. Although opportunities to quickly assess gene functions to determine their relevance to guanidine biology are currently limited, the CA8 protein from humans increases guanidine-mediated reporter gene expression in surrogate *B. subtilis* cells, which is consistent with our hypothesis that metabolic enzymes in vertebrates also manipulate guanidine concentrations. Most other genes associated with candidate guanidine-I and guanidine-II riboswitch aptamers are either directly or indirectly relevant to Ca²⁺ transport or signaling, suggesting that vertebrate guanidine riboswitches are relevant to Ca²⁺ biology.

This broad hypothesis regarding the natural role of guanidine in vertebrates is consistent with many previous observations. The effects of guanidine on neuromuscular functions were initially reported[27–30] in 1876, and its connection to Ca²⁺ signaling of human motor nerve function was observed nearly 100 years later[31]. As noted above,

guanidine has been used as a therapeutic treatment for certain neuromuscular disorders such as Lambert-Eaton myasthenic syndrome[31–33]. In addition, evidence that guanidine can overcome neurological blocks caused by neurotoxins such as curare or botulinum protein has been reported over many decades[86–89]. These previous findings are consistent with the fact that genetic disruption of the *CA8* gene associated with a guanidine-II aptamer causes the neuromuscular disorder spinocerebellar ataxia[56–58].

Both Lambert-Eaton myasthenic syndrome[90,91] and spinocerebellar ataxia[92,93] involve a mechanism of disrupted Ca²⁺ signaling. Indeed, the only proven function of CA8 protein is to block the binding of phosphatidylinositol triphosphate from its receptor ITPR1[78]. Thus, the long but sparse history of guanidine effects on vertebrates, and the previously known relationships between guanidine, CA8 protein, and Ca²⁺, are consistent with the striking associations between other guanidine-I and guanidine-II aptamer candidates and genes relevant to Ca²⁺ transport and neuromuscular functions.

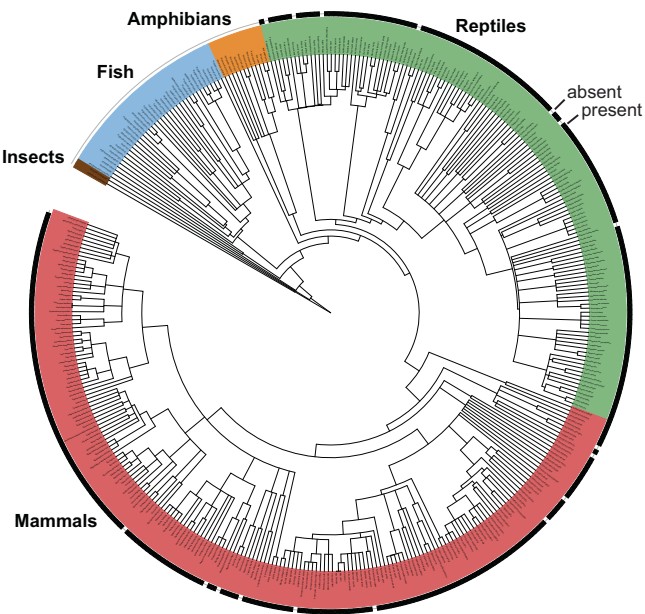

**Fig. 6 | Phylogenetic distribution of RNAs similar to guanidine-I riboswitch aptamers of bacteria in the *ITPR1* gene of vertebrate species.** A box indicates the presence of a region similar to guanidine-I aptamers for the species evaluated. A sequence alignment of the top 51 representatives is presented in Supplementary Fig. 14, including the genus and species names for their host organisms.

These correlations are consistent with the hypothesis that vertebrates exploit guanidine or a related chemical derivative to manipulate $Ca^{2+}$ signaling and utilization. If true, the locations of guanidine-I and guanidine-II aptamers in the genomes of vertebrates will reveal those genes whose functions are relevant to the nexus between the newly recognized signaling molecule guanidine and $Ca^{2+}$ signaling/utilization. Various other associated genes are involved in neuromuscular disorders or bone development diseases, which provide newfound links between these widespread human ailments and guanidine. It is intriguing to consider some of the other apparent gene associations as well. For example, the affiliation of guanidine-II aptamers with the gene for nitric oxide synthase[94,95] (*NOS1*) (Supplementary Fig. 2) presents an intriguing possibility that guanidine is relevant to the biosynthesis of the important signaling compound nitric oxide.

The most likely function of guanidine aptamers in vertebrates is in service as the sensory domains of riboswitches. The diverse locations of guanidine aptamer candidates in or near associated genes suggest that they regulate gene expression using various different mechanisms. Alternative splicing control[44–51] is commonly observed for TPP riboswitches of fungi and plants, and many vertebrate aptamer candidates are likewise located in the introns of mRNAs where they could trigger alternative RNA folding to control spliceosome access to exon splice sites or intron branch-site A regions. In rare instances, bacterial riboswitches regulate the production of antisense transcripts to control the expression of sense-orientation coding regions[96–98]. Given that eukaryotic genes are known to be regulated by various antisense mechanisms[99], the guanidine aptamer candidates in vertebrates antisense to their associated gene might exploit these same gene regulation mechanisms. Experimental validation studies will need to address numerous possible mechanisms for how ligand-binding RNAs could regulate gene expression in vertebrates.

## Methods
### Chemicals and biochemicals
Chemicals, including guanidine hydrochloride (≥99%), were obtained from Sigma-Aldrich unless otherwise noted. Aqueous solutions were prepared using deionized water (dH₂O, Milli-Q) rendered sterile using an autoclave. Radiolabeled nucleotides [γ–³²P]-ATP were obtained from Revvity and typically used within 15 days of receipt. Restriction and ligase enzymes were purchased from New England Biolabs. RNase T1 and rAPid alkaline phosphatase enzymes were purchased from Roche. Synthetic oligonucleotides (Supplementary Table 1) used for in-line probing were purchased from Sigma-Aldrich, IDT, or the Keck Oligonucleotide Synthesis facility at Yale University. A gene block containing the human *CA8* DNA sequence (Supplementary Table 1) was obtained from GenScript and used for PCR amplification and genetic transformations to generate *B. subtilis* strains carrying human *CA8* and *CA8* mutant constructs.

### Bioinformatics analyses
The initial search for homologs of guanidine riboswitch aptamers in vertebrates was conducted using the Infernal package[39] release 1.1.4. The covariation model for the previously published bacterial riboswitch aptamer classes guanidine-I[2], guanidine-II[3], guanidine-III[4], and guanidine-IV[6] was queried to search the vertebrates taxonomic divisions of the Reference Sequence (RefSeq) database[100] release 214. Hits were inspected to assess their relevance, based on their similarities to the bacterial aptamer consensus and genomic contexts. Rfam hits for guanidine-II aptamer candidates in vertebrates were identified using the full alignment of Rfam family RF01068.

Iterative searches for riboswitch aptamer candidates in vertebrates with specific gene contexts were conducted using the NCBI Entrez Gene and Nucleotide databases[101]. The process was initiated by collecting the nucleotide sequences for all vertebrate examples of a gene, wherein some contained hits of interest from the initial search. The Infernal package[39] release 1.1.4 was then used again to scan the gene sequences for additional hits of the motif by comparison to the bacterial consensus model. A revised consensus model was then generated based on all vertebrate representatives of the candidate aptamer in this context, and this consensus was used to search the gene sequences again. This process was iterated until reaching a convergence for the consensus model, and when no additional hits were identified.

### RNA constructs
RNA constructs were prepared by in vitro transcription using T7 RNA polymerase and specific synthetic DNA templates (Supplementary Table 1). The resulting RNA transcripts were separated by employing denaturing 10% polyacrylamide gel electrophoresis (PAGE). Bands corresponding to the desired RNAs were excised, and the RNAs were eluted with 500 μL crush-soak solution (200 mM NaCl, 10 mM Tris-HCl [pH 7.5 at ~21 °C], and 1 mM EDTA [pH 8.0 at ~21 °C]). The eluted RNA was precipitated using ethanol, pelleted by centrifugation at 14,000 rpm for 20 min, and the resulting pellet was resuspended dH₂O. RNAs were dephosphorylated using rAPid alkaline phosphatase (Roche), and 5 pmol of the dephosphorylated RNA was 5′-radiolabeled using (New England Biolabs) T4 polynucleotide kinase and 20 μCi [γ–³²P]-ATP in 20 μL reactions containing 25 mM CHES (pH 9.0 at ~21 °C), 5 mM MgCl₂, 3 mM DTT. The radiolabeled RNA was further purified by denaturing 10% PAGE as previously described. The sequences of the final products were partially evaluated through the results observed in the marker lanes ('OH and RNase T1 partial digestions) for in-line probing assays.

### In-line probing assays
In-line probing assays were conducted on trace amounts of 5′ ³²P-labeled RNAs using a protocol similar to that described previously[66]. Specifically, RNA was incubated in a solution containing 20 mM MgCl₂, 100 mM KCl, 50 mM Tris (pH 8.3 at ~21 °C) in the absence or presence of guanidine or an analog at concentrations specified for each assay. Reactions were incubated at room

temperature for 48 h. RNA cleavage products were resolved using denaturing 10% PAGE and visualized with a Typhoon FLA 9500 Molecular Scanner (GE Healthcare). $K_D$ values for RNA-ligand interactions were determined based on changes in cleavage product band intensities at sites designated for each analysis. Band intensities were measured using ImageJ software and $K_D$ values were calculated by curve fitting using GraphPad Prism.

## Bacterial strain preparation

Previously reported WT *B. subtilis* strain 168 1A1 or its Δ*ykkCD* variant carrying a construct with a guanidine-I riboswitch fused to a *lacZ* reporter gene[2] were made to carry either the human *CA8* gene sequence or a *CA8* mutant (here designated *CA8\**) coding for a naturally occurring Y218C amino acid change linked to spinocerebellar ataxia[74]. The guanidine-responsive reporter construct is under the control of a constitutive *lysC* promoter, and therefore reporter gene expression changes should be regulated only by the riboswitch.

The *CA8* gene was delivered to the reporter strain using the plasmid pDG1664 (accession ECE117, Bacillus Genetic Stock Center [BGSC], The Ohio State University). Insertion of the *CA8* sequence into pDG1664 was performed via restriction digestion of the initial DNA constructs with *EcoRI* and *BamHI*, gel purification of the cleaved DNAs, followed by ligation of the components using T4 DNA ligase. pDG1664 plasmids containing *CA8* were then transformed into chemically competent NEB 5-alpha *E. coli* and selected on LB agar plates containing $100\,\mu g\,mL^{-1}$ carbenicillin. The desired modified plasmid was purified from a culture initiated by a single colony using a plasmid isolation kit (QIAprep Spin Miniprep Kit, QIAGEN) following the manufacturer's directions. The purified plasmid was then used to transform both *WT* and Δ*ykkCD B. subtilis* strains carrying the reporter construct. The *CA8* sequence undergoes homologous recombination to insert at the *thrC* locus. The resulting endogenously expressing CA8 strains were isolated using culture plates containing erythromycin and chloramphenicol. Confirmation of the presence of *CA8* in the resulting strains was achieved by sequencing a PCR product of the insertion.

The *CA8* Y218C mutant strains were generated by site-directed mutagenesis of pDG1664 carrying the *CA8* gene using a QuikChange Lightning Site-Directed Mutagenesis Kit (Agilent) and the appropriate primers (Supplementary Table 1). The plasmid carrying the *CA8* Y218C mutant sequence was transformed into WT and Δ*ykkCD B. subtilis* cells as described above to yield strains endogenously expressing the CA8 Y218C mutant protein from the chromosome. Confirmation of the presence of *CA8* Y218C mutant in the resulting strains was achieved by sequencing a PCR product of the insertion.

## Riboswitch reporter assays

Liquid-based reporter assays were carried out by culturing the relevant cells overnight 30 °C in LB. The media of all strains were supplemented with $5\,\mu g\,mL^{-1}$ chloramphenicol. The media for strains carrying *CA8* or *CA8\** genes also included $0.5\,\mu g\,mL^{-1}$ erythromycin. Overnight cultures were diluted 1:50 into fresh LB supplemented with the appropriate antibiotics in the absence or presence of $300\,\mu M$ guanidine and incubated at 37 °C either for 6 h or until an $OD_{600}$ of ~1.2, as indicated for each experiment. The resulting cells were subjected to quantitative β-galactosidase Miller assays[102] using *o*-Nitrophenyl β-D-galactopyranoside (ONPG) as the substrate.

Agar diffusion assay using various *B. subtilis* strains exposed to guanidine was conducted with solid LB media including $100\,\mu g\,mL^{-1}$ X-gal and the appropriate antibiotics as described above. Ten mL of 6 M guanidine was applied to a cellulose filter disc, and the development of blue color was evaluated after 24 h incubation at 37 °C[2,53]. The experiments were performed in duplicate or triplicate, and representative photographs are presented and quantitated. Comparison of β-galactosidase activities for different strains examined by agar diffusion assays was conducted by plotting the intensities (in arbitrary units) of pixels on a line through the center of the filter discs in photographs measured with ImageJ software.

## Reporting summary

Further information on research design is available in the Nature Portfolio Reporting Summary linked to this article.

## Data availability

All data needed to evaluate the conclusions herein are presented in the main or Supplementary Information sections. Source data are provided with this paper.

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

## Acknowledgements

The authors thank Seth Lyon for contributing to the assessment of the function of human CA8 protein, Gabriel Belem de Andrade for alerting us to the Rfam hits, and other members of the Breaker laboratory for helpful discussions. M.G.M. was supported by the Howard Hughes Medical Institute via a postdoctoral fellowship from the Life Sciences Research Foundation. We particularly thank the G. Harold & Leila Y. Mathers Foundation who provided early financial support for this high-risk project. This work was also supported by funds from the Howard Hughes Medical Institute and Yale University.

## Author contributions

The original project concept was formulated by K.K. and R.R.B. Bioinformatics analyses were planned and executed by I.M. and A.N. Biochemical and genetic experiments were planned or executed by K.K., J.J.M., M.G.M., Y.C. and R.R.B. R.R.B. wrote the manuscript with input and edits from all authors. Project management and primary funding acquisition were carried out by R.R.B.

## Competing interests

The authors declare no competing interests.
