## [Transparent Peer Review file · Nature Communications]

Guanidine aptamers are present in vertebrate RNAs associated with calcium signaling and neuromuscular function

Corresponding Author: Professor Ronald Breaker

Version 0:

Reviewer comments:

Reviewer #1

(Remarks to the Author)

Summary:

In this study, the authors discovered that guanidine is sensed by the aptamer of the guanidine riboswitch. These aptamers, found in vertebrates, are located in the same region of genes that play a crucial role in Ca²⁺ signaling and, consequently, in the neuromuscular function. These findings are supported by bioinformatics analysis, in-line probing assays, and agar diffusion assays. First, they identified guanidine-II riboswitch aptamer candidates in the antisense strand of the CA8 gene, using their own analysis and the Rfam database. Second, they show that the aptamer can selectively bind to guanidine in vitro by forming short tandem hairpins that cooperatively bind to the ligand. Also, they show that loops H5, H6, and H7 of the 112 CA8 construct from the African elephant undergoes structural changes when guanidine concentrations are elevated, thus indicating the cooperative binding of guanidine to these sites and the interaction between hairpins. Third, they demonstrate that the human CA8 protein increases guanidine concentration when expressed in a *B. subtilis* strain carrying a guanidine-responsive reporter gene. Finally, they claim that the guanidine-II aptamer candidates in vertebrates are associated with genes important to Ca²⁺ biology, like the PISD and CACNA1C genes, using Rfam database. They also measure the KD of the corresponding RNA aptamer constructs for the guanidine.

Overall, I think the article is interesting and requires several modifications before being considered by Nature Communications.

Article

Major comments:

1- This manuscript is about the identification of novel aptamers in vertebrates and their potential implication in a riboswitch regulation system. However, in the first case (CA8), the aptamer is found in the antisense strand, which makes difficult to understand how it could regulate the expression of CA8. In the second case (PISD), the authors mention that the aptamer is found upstream (~10 kb) of the gene but are not sure if it is in the same RNA molecule, thus making hard to judge whether the aptamer could be involved in the regulation. Therefore, in both cases, it is critical for the authors to include a paragraph explaining if they think that the guanidine aptamers are related to a riboswitch regulation mechanism. If so, it should be discussed based on the possible mechanisms involving the guanidine riboswitch. Basically, what is the role of the riboswitch and how does it work? Based on the current experimental evidence, it could be possible that the authors have found RNA aptamers that only perform guanidine binding but that do not regulate gene expression, i.e., riboswitch remnants that are no longer functional. If the authors do not prefer to commit themselves to the riboswitch hypothesis, they should discuss their current position about what their finding represents (riboswitch remnants?) and possible perspectives of the research.

2- The authors showed that the human CA8 protein increases guanidine concentration when expressed in *B. subtilis* using agar diffusion assay. However, there is no evidence that their findings are applicable in human cells. To ascertain the function of CA8, it is critical that it is tested in human cells. Without such data, it is hard to conclude about the function of CA8. Interestingly, the same team has previously shown the function of the TPP riboswitch in plants (Cheah et al., Nature 2007) using in vivo assays in *Arabidopsis thaliana*, indicating that they know how to handle such assays.

3- There are some KD values with no error values, like for Figures 6, 7, 8 and 10, in the supplementary material. It is not known if the authors did the experiment as triplicates or not. It is important to show that the results obtained in an experiment

are replicable and biologically relevant. It is thus important to state how many replicates were performed for these experiments and add the standard deviation for each measurement.

Minor comments:

Line 90: Remove the parentheses in "(in rare instances)".

Line 95: "than is currently known." → "than it is currently known".

Line 126: Remove "(not always)".

Line 134: PSD should be in italic (gene name). Check other occurrences.

Line 170: "that is like that" → similar

Line 202: "... that the nine hairpins that mimic guanidine-II binding aptamers..." → that the nine hairpins, which mimic guanidine-II binding aptamers, ..."

Line 314: *Rhinoptilus africanus* should be in italic.

Figure 1: It seems too cramped and would benefit by adding more space between the panels.

Figure 2c: It is hard to discern the different symbols because they are close to each other in the graph. I suggest using a white filling with different outline colors for each series.

Figure 3c: Same comment as for Figure 2c.

Figure 3d: Same comment as for Figure 2c. Also, please describe the use of asterisks in the legend. I believe it is for the Y218C mutation, but it should be explicitly mentioned in the legend.

Supplementary material:

Minor comments:

Figure 3b: Same comment as for Figure 2c.

Figure 3d: Same comment as for Figure 2c.

Line 99: 112-nucleotide → 112-nucleotide

Line 169: CA8 should be in italic (name of a gene).

Table S1. For the oligos "Dolphin-2F" and "Dolphin-2R", I think there is a mistake in the annotation: "for with downstream".

Reviewer #2

(Remarks to the Author)

Reviewer #3

(Remarks to the Author)

This paper by Kavita et al. presents a significant discovery—the presence of guanidine-sensing riboswitch aptamers in vertebrate genomes, challenging the long-standing paradigm that metabolite-sensing riboswitches are largely restricted to bacteria. Through comparative sequence analysis, covariance modeling, and functional validation, the authors build a compelling case that vertebrates utilize guanidine-responsive RNA elements to regulate genes associated with calcium signaling and neuromuscular function. This manuscript presents a novel conceptual advance with broad relevance and strong methodological rigor, making a valuable contribution to the field of RNA biology and gene regulation. However, there are a few comments and clarifications that should be addressed to improve the manuscript further.

Comments:

1. The manuscript suggests that guanidine-responsive riboswitches are widespread in vertebrates, yet the evolutionary origin of these aptamers is not deeply discussed. Are these aptamers conserved across orthologous loci in multiple vertebrate species?
2. Could the authors provide a phylogenetic alignment of representative hits (e.g., in CA8 and ITPR1) to evaluate conservation?
3. Do the authors consider these aptamers to have evolved convergently in eukaryotes, or is there any evidence suggesting horizontal gene transfer or ancient vertical inheritance?
4. While this study provides evidence for guanidine-sensing riboswitches in vertebrates, it would be valuable to explore whether structure-guided methods, such as inverse RNA folding, could uncover additional aptamers with conserved structures but low sequence similarity. This could broaden our understanding of RNA-based regulation and its evolutionary diversity.
5. Can the authors provide structure-annotated multiple alignments across vertebrate orthologs to assess base-pair covariation and support structural conservation beyond primary sequence similarity? This would strengthen the evidence for evolutionary and functional conservation of the identified aptamers.
6. The existence of multiple guanidine-like hairpins (e.g., the nine in elephant CA8) raises interesting questions. Could the authors clarify whether they expect these hairpins to work together in a cooperative way, beyond just acting in pairs? Is there any possibility of higher-order structural interactions or coordinated binding?
7. The authors note the simplicity of the guanidine-II aptamer motif, which could increase susceptibility to false positives. Was any background model tested (e.g., scanning randomized sequences or intronic sequences from unrelated genes) to quantify the false discovery rate?

Reviewer #4

(Remarks to the Author)

The paper "Widespread vertebrate aptamers reveal a major role for guanidine in regulating calcium signaling and neuromuscular function" discovered new guanidine-I and guanidine-II riboswitch aptamer candidates in a wide range of vertebrate genomes, with comprehensive biochemical validation. The authors used computational algorithms to identify new candidates for guanidine-I and guanidine-II riboswitch aptamers from vertebrate genomes, including birds, elephants, dolphins, and fish et al. They employed in-line probing to detect structural changes in the guanidine-II candidate constructs in elephants as guanidine concentration varied, providing strong evidence for the guanidine-II riboswitch aptamer in vertebrates. Additionally, the multiple sequence alignment and gene localization analysis related to CA8 and genes relevant to Ca²⁺ biology are very convincing. The discovery of riboswitches in vertebrates is a surprising scientific finding; although there are only two types of guanidine in this paper, it is enlightening and suggests that riboswitch regulation may still be widespread in vertebrates. I would like to suggest the following points for improvement.

Minor comments

1. The section "Evidence that the human CA8 protein increases guanidine concentration when expressed in 225 bacteria" gives evidence about the function of CA8 protein on increases guanidine concentration. But it is not direct evidence for the existence of guanidine riboswitch aptamer in vertebrates. A couple of reasoning sentences may better explain the logic.
2. In the section "Numerous vertebrate RNAs resemble portions of the consensus model for the bacterial guanidine-I aptamer class". The experimental evidence (supporting the existence of guanidine-I in vertebrates) is less than evidence for guanidine-II aptamer candidates.
3. In line 261, "300 μ M guanine" could be a typo of "300 μ M guanidine."?

Version 1:

Reviewer comments:

Reviewer #1

(Remarks to the Author)

Point 1: We agree with the authors clarification that the study presents evidence that, in vertebrates, guanidine influences muscle function via guanidine aptamers located near genes important to Ca²⁺ signaling and neuromuscular function.

Point 2: We understand why assays for CA8 function were not carried out in human cells, given the absence of a known genetic reporter system to monitor guanidine in eukaryotic cells and the lack of human cell culture experience in the laboratory.

Point 3: We thank the authors for providing a context about the KD measurements that they have done in the past. However, the KD values in the manuscript (lines 196, 211, and 310) still lack error values, which are important for determining the reliability and reproducibility of binding measurements. KD values with no error values prevent evaluation of statistical significance and limits the comparison of results with other studies. For these measurements to be acceptable, we consider that the error values must present. In this particular paper, the significance of KD values is highly important to validate the presence of these new guanidine aptamers.

In addition, in line 196, the KD is 110 μ M but it is shown as 115 μ M in Figure 2. This apparent discrepancy needs to be clarified.

Finally, we realised that the KD values mentioned in the text are not presented in any figures (in the article nor in the supplementary data), making it difficult to verify or interpret the results. The data supporting these numbers should be added to the text.

Point 4: We understand if the authors prefer to reuse the traditional symbol markings, commonly used in some journals. We appreciate that the authors included an Excel sheet with the values for each point in all plots, as this can aid in data interpretation.

Reviewer #2

(Remarks to the Author)

Reviewer #3

(Remarks to the Author)

The authors have addressed all the reviewer comments and concerns in the revised manuscript.

Reviewer #4

(Remarks to the Author)

The authors have completely addressed all my concerns.

Reviewer #5

(Remarks to the Author)

I was invited to serve as an adjudicating referee on the specific question of KD measurements and their presentation in this study. In my view, the primary value and major finding of the paper lie in the identification of guanine riboswitches in vertebrates. This conclusion is unequivocally supported by the data presented and discussed, including Figure panels 2b and 2c. The data clearly demonstrate that the KD values of these riboswitches in vertebrates fall within the 100 μ M range.

In my opinion, the precise KD value—whether it is 300 μ M, 200 μ M, 100 μ M or even 50 or 10 μ M—is not critical to the significance of this work. The key finding is the existence of these riboswitches in vertebrates, which represents a major contribution to the field.

I am therefore satisfied with Figure 2 in the main text, which presents a representative sequence example, an in-line probing gel, and the extracted KD value annotated as $115 \pm 8 \mu$ M, based on three replicates. I have reviewed the manuscript and the response to reviewers (RTR), and I find the authors' explanation to be thorough and satisfactory.

While I acknowledge that in-line probing may not be the most precise method for determining KD values, this does not detract from the significance of the findings. It leaves room for further research, and I am confident that other groups will refine these measurements using more precise techniques, such as isothermal titration calorimetry (ITC), microscale thermophoresis (MST), 2-aminopurine fluorescence spectroscopy, or simple NMR spectroscopic methods using ¹⁵N-labeled guanidine.

Responses to Reviewer Comments

NCOMMS-25-19555

Title: *Widespread vertebrate aptamers reveal a major role for guanidine in regulating calcium signaling and neuromuscular function*

Authors: Kavita et al. and Breaker

Reviewer 1

Point 1: The reviewer requests additional information on whether we think the guanidine aptamers are components of riboswitches and how these riboswitches work.

Response: Although we do not have direct experimental proof of gene control function, all the various datasets reported in our manuscript support the hypothesis that guanidine aptamers in vertebrates indeed are parts of riboswitches. This view was expressed in the Title, Abstract, Introduction, and Concluding Remarks section of the original manuscript. We are always careful to state that these are “riboswitch candidates”, but this is only exercising extreme caution in the event that a few of the many representatives might be genetically defective, or if some aptamers have some other purpose apart from gene regulation.

The issue of vertebrate riboswitch mechanisms is far more challenging to address at this early stage. In bacteria, there are rare examples of riboswitches that control a gene by regulating the production of antisense transcripts, and we believe aptamers in antisense transcripts will be used for gene control in vertebrates too. In addition, we have other (unpublished) riboswitch classes in mammals that are also often in the antisense orientation, indicating that antisense representatives will be common.

Although we would prefer to have experimental details regarding how these riboswitches function, we believe this larger story will play out in stages much like it did for eukaryotic TPP riboswitches. For the previously discovered TPP riboswitches in fungi and plants, we and others published on the existence of the aptamers several years before making the difficult formal proofs of gene control mechanisms. We think our current findings are too important to hold for several years until we obtain the funding and generate the data for the precise mechanisms use by vertebrate guanidine riboswitches. At this time, to emphasize the plausibility of various gene regulation mechanisms, we have added the following paragraph to the Discussion section of the revised manuscript, including the addition of our more citations:

“The most likely function of guanidine aptamers in vertebrates is in service as the sensory domains of riboswitches. The diverse locations of guanidine aptamer candidates in or near associated genes suggests that they regulate gene expression using various different mechanisms. Alternative splicing control⁴⁴⁻⁵¹ is commonly observed for TPP riboswitches of fungi and plants, and many vertebrate aptamer candidates are likewise located in the introns of mRNAs where they could trigger alternative RNA folding to control spliceosome access to exon splice sites or intron branch-site A regions. In rare instances, bacterial riboswitches regulate the production of antisense transcripts to control the expression of sense-orientation coding regions⁹⁶⁻⁹⁸. Given that eukaryotic genes are known to be regulated by various antisense mechanisms⁹⁹, the guanidine

aptamer candidates in vertebrates antisense to their associated gene might exploit these same gene regulation mechanisms. Experimental validation studies will need to address numerous possible mechanisms for how ligand-binding RNAs could regulate gene expression in vertebrates.”

On the topic of genetic “remnants”, the discussion can become much more complicated, and we have intentionally avoided such a discussion in the current manuscript. In bacteria, we never convincingly find a riboswitch remnant because these species are often under extreme evolutionary pressure to eliminate junk sequences from their genomes. In contrast, eukaryotic species are far more tolerant of disrupted genetic components and can carry them for much longer periods. However, given the genetic distributions, conserved sequences, and conserved secondary structures, we believe that the vast majority of the vertebrate aptamers are functional. Unfortunately, without far more experimental analyses, including bioinformatics, statistics, biochemical spot checking, and gene control assays, we are not well positioned to make a definitive statement about the evolution of these RNAs.

In summary, we feel that the original manuscript properly presents the evidence that vertebrates carry many guanidine aptamers in or near genes relevant to Ca^{2+} signaling and neuromuscular function. Our evidence indicates that many will prove to function as guanidine riboswitches, although we believe they should be considered “riboswitch candidates” until additional direct analyses of specific candidates is completed in future studies. We believe that the original draft of the manuscript presents the new data with sufficient caution while also helping to solve a 150-year-old mystery of why guanidine influences muscle function. Therefore, we would prefer not to make additional major edits in response to this point. See also the response to Point 2 of Reviewer 4.

Point 2: The reviewer states that assays for CA8 function should be carried out in human cells.

Response: As we note in the original manuscript, we cannot conduct the definitive experiment where purified CA8 protein is provided substrate candidates to look for enzymatic activity because the most likely substrate is not chemically stable. The second-best experiment would be to test CA8 activity in human cell culture (or perhaps some other vertebrate cell line). However, there is no known genetic reporter system to monitor guanidine in eukaryotic cells. Moreover, we do not have human cell culture experience in the laboratory. The reviewer also notes our ~18-year-old work with fungal or plant TPP riboswitches, but these experiments were for studying the mechanisms of riboswitches, not for studying the functions of proteins of unknown function like CA8. We would need to spend years engineering a eukaryotic genetic reporter system for guanidine or make use of more traditional chemical methods for measuring in vivo guanidine concentrations (which are often not very quantitative).

Without an abundance of new funding and unlimited time, eukaryotic cell testing of CA8 activity is simply impractical. Therefore, we decided to exploit a bacterial system for cell-based testing of CA8 function because of the power of the guanidine-responsive bacterial genetic reporter system we created some years ago. Instead of delaying publication for years, we would prefer to publish our findings on CA8 as described in the original manuscript.

Point 3: The reviewer requests clarification on the reproducibility of binding assays and the error values on reported K_D numbers.

Response: My laboratory created the ‘in-line’ probing assay for establishing K_D values for aptamers. Therefore, we have faced this question of reproducibility for the many dozens of riboswitch papers we have published that make use of this method. Through this extensive experience, we find that the assay is highly reproducible, and yields K_D values that are typically within two-fold between runs, even when replicated using independent material or when different experimenters conduct the assays. Moreover, we are always cautious when interpreting these results so as not to make profound conclusions based on tiny differences in K_D values.

For the current study, K_D values derived from data depicted in main text figures have been reproduced in triplicate as described in the original manuscript. Other K_D values discussed are derived from representative results depicted in the Supplementary Information section. These results are often derived after scout reactions (not shown) were conducted to identify the approximate K_D values, which serve as informal experimental repeats. Moreover, within each in-line probing reaction series, multiple reaction mixtures using various concentrations of guanidine are prepared and evaluated, and thus each K_D assay in the Supplementary Information section is actually a series of reaction repeats at slightly different ligand concentrations. Although each full reaction series was not reproduced in triplicate like the data depicted in **Fig. 2c** and **Supplementary Fig. 3**, remarks about the differences in K_D values for the constructs discussed involve differences as great as 20-fold – which is far outside the range of typical experimental errors.

In addition, our K_D values reported in many past manuscripts are often confirmed by other researchers who use their favorite method for establishing affinities (ITC, SHAPE, MST, etc...). Given the general reproducibility of K_D values established by in-line probing, given that we make no major conclusions based on tiny differences in K_D values, and given that conducting in-line probing assays is expensive, time consuming, and uses radioactive tracers, we would prefer to publish the existing data sets. However, in the revised manuscript, we now clarify that some of the K_D values are based on single representative assays presented in the Supplementary Information section.

Point 4: The reviewer lists 16 minor comments.

Response: We have considered all minor recommendations and made adjustments in most instances accordingly. However, we did not change the symbol markings for plots from the traditional series used by some journals (filled circles, open circles, filled squares, etc.). In all instances, we feel the results are readily interpretable and, in some cases, the different points indicate simple replicates such that there is no need to perfectly distinguish the points. In addition, we now include an excel spreadsheet with the values for each point in all plots.

Reviewer 2

This was a co-reviewer, presumably with Reviewer 1. There were no separate criticisms to address.

Reviewer 3

Point 1: The reviewer notes that we do not discuss the evolutionary origin of the vertebrate guanidine aptamers, and asks “Are these aptamers conserved across orthologous loci in multiple vertebrate species?”.

Response: We mostly try to avoid discussion of the evolution of riboswitches or their components (except for general statements about the RNA World) because it is very difficult to determine origins of these RNAs with confidence. Fortunately, addressing such questions (interesting as they are!) is not necessary to support our main conclusion that vertebrates carry many guanidine aptamers in or near genes relevant to Ca^{2+} signaling and neuromuscular function.

Although there are many genes associated with guanidine riboswitch candidates in vertebrates, we have only comprehensively look at orthologous loci for the guanidine-II candidate associated with the *CA8* gene and the guanidine-I candidate associated with the *ITPRI* gene using methods described in the original manuscript. The *CA8* guanidine-II candidate is rare and therefore no phylogenetic tree was created. The *ITPRI* guanidine-II candidate is readily detectable in most reptilian and mammalian species, but is not apparent in the *ITPRI* gene of fish or amphibians. We had planned to use this data in a future manuscript on vertebrate guanidine-I riboswitches, but we now include this data as **Fig. 5** of the revised manuscript. The manuscript has been edited accordingly. See also the response to Point 3 of Reviewer 3.

Point 2: The reviewer asks if we can provide an alignment for representative *CA8* and *ITPRI* hits.

Response: We are a bit uncertain if the reviewer is asking about sequence alignments or phylogenetic distributions, and therefore we address both topics here. The sequence alignments were presented in the original manuscript draft as **Supplementary Fig. 1** (*CA8*) and **Supplementary Fig. 14** (*ITPRI*). Slight edits to the call-ins for **Fig. 4** and its associated supplementary figures will help make this clearer. Regarding the phylogenetic distribution of candidates, as noted in the response to Point 1 of Reviewer 1, we have also added the phylogenetic tree for the *ITPRI* candidates (**Fig. 5**), which reveals widespread conservation of the guanidine-I-like motif in reptiles and mammals.

Please also note that we suspect that many more vertebrate *CA8* genes carry guanidine riboswitches based on a single hairpin of the guanidine-II type, perhaps supported by another non-consensus hairpin (as is the case for elephant H4 and H5) or a different substructure. However, this hypothesis has not been explored. This potential sequence and structural diversity for the *CA8* RNAs makes it much harder to create a comprehensive set of hits and alignments, and therefore we restricted our discussion in the current manuscript to the *CA8* candidates that exhibit tandem hairpins that conform to the guanidine-II consensus observed in bacteria.

Point 3: The reviewer asks a question about the evolutionary origin of vertebrate guanidine aptamers.

Response: Guanidine-II riboswitch aptamers are very simple, and therefore could emerge spontaneously quite easily. This might be the explanation for some examples in vertebrates because the P1 and P2 regions appear to be identical or nearly identical, suggesting a recent duplication of an existing hairpin served as their origin. However, this mechanism of emergence

cannot be true for all examples we observe due to the complexity of guanidine-I candidates and some of the unusual features compared to their bacterial counterparts. Again, we would prefer not to speculate in detail on the evolutionary histories of these RNAs, as this is a difficult area to generate solid conclusions. See also the response to Point 1 of Reviewer 3.

Point 4: The reviewer suggests another way that could be used to effectively search for eukaryotic riboswitches generally conforming to the known structural classes.

Response: In our study of guanidine riboswitch candidates in vertebrates, as with our study of other riboswitch classes in mammals (unpublished work), we are finding that our consensus models based on bacterial riboswitch aptamers are too conservative. The eukaryotic examples often carry greater differences or are otherwise more tolerant of mismatches to the bacterial consensus models – all while retaining their functions. The approach recommended by the reviewer would be ideal to further expand the number of riboswitch candidates. We don't think the reviewer is asking for edits to our current manuscript regarding this point, but the approach suggested would indeed be useful for future studies.

Point 5: The reviewer is requesting annotated versions of sequence alignments to assist in evaluation of structure conservation.

Response: These were already presented as **Supplementary Fig. 1 (CA8)** and **Supplementary Fig. 14 (ITPRI)** of the original manuscript. The utility of these files is best observed for the guanidine-I representatives in the *ITPRI* gene as depicted in the consensus in **Fig. 4c** and in **Supplementary Fig. 14** of the original manuscript. Although some base-pairing mismatches are observed (as indicated by the lack of shading), evidence for both covariation and compatible mutations exists for P1 and P2 are evident on careful visual inspection.

Point 6: The reviewer asks if we think the multiple pairs of hairpins in the elephant *CA8* gene “work together in a cooperative way”.

Response: Yes! We believe that our guanidine binding assays with tandem hairpins (**Fig. 2**) and with four hairpins (**Supplementary Fig. 6** and **Supplementary Fig. 7**) demonstrate that there must be cooperative interplay both between tandem hairpins and between groups of tandem hairpins. We make our views on this topic clear in the original manuscript in the section now entitled “A cluster of guanidine...”.

Point 7: The reviewer asks if we have conducted testing to determine the rate of false positive hits from the guanidine-II search.

Response: We have not formally conducted analyses to determine the expected and observed rates for false positives, but we believe these will be very low. For the guanidine-II class, our findings are similar to that observed from the Rfam database hit list, where the vast majority of apparent hits reside in or near genes relevant to Ca^{2+} biology or neuromuscular function (**Supplementary Fig. 2**). This strong clustering of gene associations indicates these hits are mostly legitimate. Likewise, this same pattern is observed for the guanidine-II candidates. We

think these points are evident in the original draft of the manuscript and therefore we have not made additional edits regarding this point.

Reviewer 4

Point 1: The reviewer recommends some clarifying words be added to a section of the manuscript on CA8 assays.

Response: Because guanidine riboswitches in bacteria led us to discover genes relevant to guanidine transport and metabolism in bacteria, we suspected that CA8 might therefore have a related function in vertebrates. We have added the following text to the relevant section that gives a reason for why we believed CA8 is relevant to guanidine biology:

“Guanidine riboswitches in bacteria regulate the expression of proteins that transport or metabolize this same ligand.”

Point 2: The reviewer notes that we have less experimental evidence for guanidine-I aptamers than for guanidine-II aptamers.

Response: As we note in the original manuscript, we do not have classical biochemical or genetic data supporting the function of the guanidine-I aptamer candidates. A key problem we encountered is that the RNA we tested is not well folded (an occasional problem even when studying bacterial aptamers outside their native environments). However, our bioinformatics results, including gene associations of the hits, strongly indicate that these RNAs are likely components of guanidine riboswitches. We are now working to secure funding to pursue formal proofs of riboswitch function, in a phased approach similar to what we and others did when discovering and validating the first fungal and plant TPP riboswitches. See also the response to Point 1 of Reviewer 1.

Point 3: The reviewer identified a typo in the manuscript.

Response: The typo has been corrected.

Additional Edits

1. A typo was fixed in the legend to **Supplementary Fig. 8**.
2. Edits were made to more clearly link **Fig. 4** to the main text and **Fig. 4** panels were reorganized accordingly.
3. Several other minor typos were fixed.
4. The section names were edited to conform to the manuscript style.
5. The subsection names have been revised, primarily to shorten them to fit the limit of 60 characters.
6. Various other edits have been made to conform to the journal format style. However, we have not eliminated all uses of the word “novel”. For example, we do not want to describe a new aptamer as “new” because this is scientifically incorrect. The new aptamer

is probably very ancient, but it is “new” or “novel” to us researchers – and “novel” is a perfect description of what we mean.

7. Additional information was added to address the question of reagent and sample validation.